# HU-671, a Novel Oleoyl Serine Derivative, Exhibits Enhanced Efficacy in Reversing Ovariectomy-Induced Osteoporosis and Bone Marrow Adiposity

**DOI:** 10.3390/molecules24203719

**Published:** 2019-10-16

**Authors:** Saja Baraghithy, Reem Smoum, Malka Attar-Namdar, Raphael Mechoulam, Itai Bab, Joseph Tam

**Affiliations:** 1Obesity and Metabolism Laboratory, The Institute for Drug Research, School of Pharmacy, Faculty of Medicine, The Hebrew University of Jerusalem, Jerusalem 9112001, Israel; saja.baraghithy@mail.huji.ac.il; 2Bone Laboratory, Institute for Dental Research, Faculty of Dentistry, The Hebrew University of Jerusalem, Jerusalem 9112001, Israel; reems@ekmd.huji.ac.il (R.S.);; 3Medicinal Chemistry Laboratory, The Institute for Drug Research, School of Pharmacy, Faculty of Medicine, The Hebrew University of Jerusalem, Jerusalem 9112001, Israel; raphaelm@ekmd.huji.ac.il

**Keywords:** bone lipids, *N*-acyl amide, oleoyl serine, osteoporosis, bone marrow adiposity

## Abstract

Oleoyl serine (OS), an endogenous fatty acyl amide (FAA) found in bone, has been shown to have an anti-osteoporotic effect. OS, being an amide, can be hydrolyzed in the body by amidases. Hindering its amide bond by introducing adjacent substituents has been demonstrated as a successful method for prolonging its skeletal activity. Here, we tested the therapeutic efficacy of two methylated OS derivatives, oleoyl α-methyl serine (HU-671) and 2-methyl-oleoyl serine (HU-681), in an ovariectomized mouse model for osteoporosis by utilizing combined micro-computed tomography, histomorphometry, and cell culture analyses. Our findings indicate that daily treatment for 6 weeks with OS or HU-671 completely rescues bone loss, whereas HU-681 has only a partial effect. The increased bone density was primarily due to enhanced trabecular thickness and number. Moreover, the most effective dose of HU-671 was 0.5 mg/kg/day, an order of magnitude lower than with OS. The reversal of bone loss resulted from increased bone formation and decreased bone resorption, as well as reversal of bone marrow adiposity. These results were further confirmed by determining the serum levels of osteocalcin and type 1 collagen C-terminal crosslinks, as well as demonstrating the enhanced antiadipogenic effect of HU-671. Taken together, these data suggest that methylation interferes with OS’s metabolism, thus enhancing its effects by extending its availability to its target cells.

## 1. Introduction

Bone is a remarkably intricate and metabolically active organ that serves essential functions, including providing the structural and mechanical integrity required for locomotion and protection of vital organs, maintenance of mineral homeostasis, and hematopoiesis [1,2]. The consonance between structure and functionality is maintained by a tightly coordinated remodeling process in which bone tissue is broken down by osteoclasts and rebuilt by osteoblasts. Bone remodeling is continuously governed by numerous regulators at the local and systemic levels [3,4,5,6]. However, with aging, remodeling leans towards a negative bone balance, thus leading to osteopenia and osteoporosis, consequently precipitating bone fractures [7,8,9,10].

Lipids comprise a large group of chemically diverse compounds. Recent advances in the field of lipidomics have vastly expanded our understanding of the different biological functions of lipids. It is now well established that the roles of lipids are not solely limited to energy storage or as structural components of the cell membrane [11]—they also include their activity as signaling molecules, which are important in several physiological and pathological conditions [12,13,14,15]. Our research focus for the past few years has been on the skeletal endocannabinoid system [16,17,18,19]. Endocannabinoids, their congeners, and related compounds, are fatty acid derivatives consisting of saturated and unsaturated (ω-3, ω-6, ω-7, and ω-9) long-chain fatty acid amides (FAAs) [20,21]. Although a few endogenous individual members of the FAA family have been known since the mid-20th century, it is the discovery of arachidonoyl ethanolamide (AEA) or anandamide [22] that boosted the modern, large-scale investigation of FAAs and their designation as a family. Recently, a growing body of evidence has indicated that FAAs are present in bone cells and that they play an important role in the regulation of bone mass [11,23]. Such an example is *N*-oleoyl-l-serine (OS); investigation into its metabolic activity in bone demonstrated that it is a potent anti-osteoporotic agent in both in vitro and in vivo models [24]. Recently, we have reported a role for OS in the pathological skeletal manifestations associated with Prader–Willi syndrome (PWS) [25]. Loss of *Magel2*, one of the genes in the PWS-critical region, induced a significant reduction in bone mass that was correlated with reduced circulating levels of OS both in humans and mice.

Although OS is a promising compound from a therapeutic point of view, its stability in the body is questionable, since it is an unprotected amide that can be easily hydrolyzed by circulating amidases. Hypothesizing that hindering the amide bond by adjacent methyl substituents may restrain OS’s hydrolysis and prolong its activity, as shown previously with AEA [26], we synthesized several methylated derivatives and evaluated their efficacy. Of the compounds tested, oleoyl α-methyl serine (HU-671) exhibited enhanced efficacy, both in stimulating osteoblast proliferation and activity and in inhibiting osteoclastogenesis and osteoclast activity. Furthermore, HU-671 fully attenuated the bone loss in *Magel2^−/−^* mice via a positive modulation of bone remodeling [25].

In the present study, we aimed to further investigate the skeletal effect of methylation on OS anti-osteoporotic effects. For that purpose, we evaluated the efficacy of two methylated OS derivatives regarding the reversal of ovariectomy (OVX)-induced bone loss, the most established animal model for osteoporosis.

## 2. Results

### 2.1. HU-671 Rescues Ovariectomy-Induced Bone Loss

To evaluate the efficacy of OS and its methylated derivatives (Figure 1A–C), their skeletal effects were tested on ovariectomized (OVXed) mice, an established murine model for osteoporosis. According to the schematic presentation of the experimental protocol shown in Figure 1D, sexually mature female mice were OVXed or sham-OVXed, and analyzed one week after surgery to allow for postoperative recovery and to serve as baseline controls. Five additional weeks were allowed to pass before initiating treatment with OS and its methylated derivatives in the remaining groups, in order to permit significant bone loss to occur in the OVXed animals. At that time, one group of OVXed (6W/OVX) and the corresponding sham-OVXed mice (6W/Sham/OVX) were euthanized to evaluate pretreatment bone loss. Then, daily intraperitoneal treatment with Vehicle (12W/OVX/Veh), OS (12W/OVX/OS), HU-671 (12W/OVX/HU-671), or HU-681 (12W/OVX/HU-681) was initiated for an additional 6 weeks. No adverse side effects nor negative effects on survival were noted in any of the treatment groups.

All animals were subjected to combined microcomputed tomography (µCT) and histomorphometric skeletal analyses. One week post-operation, there was no statistical difference between the bone volume densities (BV/TV) of the Sham/OVXed and the OVXed animals (Figure 2A–C). However, there was a significant bone loss at the six-week time point, indicating that the surgical procedure was successful at inducing osteoporosis (Figure 2A–C). OS and HU-671 rescued the bone loss in the femora with an insignificant difference in BV/TV between 12W/Sham-OVX/Veh animals and the treated mice (Figure 2A). The effect of HU-671 was obtained at 0.5 mg/kg/day, compared to OS at 3 mg/kg/day, consistent with a 10-fold stronger efficacy of HU-671 than with OS in ex vivo assays [25].

3D μCT images of trabecular BV/TV illustrate the notable rescue effect of OS and HU-671 on the distal femoral metaphysis (Figure 2D). This effect was secondary to a significant increase in trabecular thickness (Tb. Th.; Figure 3A), and the correction of the reduced trabecular number (Tb. N.; Figure 3B) and trabecular spacing (Tb. Sp.; Figure 3C) found in the OVX/Veh-treated animals. Interestingly, HU-681 had a lower effect on all trabecular parameters (Figure 2 and Figure 3). The rescue of OVX-induced bone loss by OS and HU-671 was selective for the trabecular bone compartment, since the increased medullary cavity diameter and decreased cortical thickness were unaffected by the treatment (Figure A1).

Since the regulation of trabecular bone mass may differ among skeletal sites, the effect of the drugs on the cancellous compartment in L3 bodies was also analyzed. Both OS and HU-671 successfully rescued trabecular bone loss in vertebral bodies. Similarly, to the effect documented in the femora, HU-671 had a peak effect at 0.5 mg/kg/day, and OS at 3 mg/kg/day, whereas HU-681 had no significant effects (Figure 4A–D).

### 2.2. HU-671 Mitigates Bone Resorption and Enhances Bone Formation

Consistent with our previous findings [21], OS and HU-671 at 3 mg/kg/day and 0.5 mg/kg/day respectively, decreased the number of TRAP-positive osteoclasts in the femora of the OVXed mice (Figure 5A,B) and normalized the serum levels of CTX-1, a marker of bone resorption (Figure 5C). These findings further support the anti-resorptive effects of these compounds. Investigating bone formation parameters by utilizing dynamic calcein labelling of newly mineralized bone fronts, as can be seen in Figure 5D, revealed that both drugs increased the bone formation rate (BFR) and the mineral appositional rate (MAR) (Figure 5D–F) in comparison with the OVX/Veh-treated animals; however, the reduction in osteocalcin serum levels of was not fully reversed in mice treated with either OS or HU-671 (Figure 5H). Moreover, HU-671 significantly increased the mineralization surface (MS/BS; Figure 5G), suggesting an enhanced skeletal anabolic activity by α-methylation of OS.

### 2.3. HU-671 Reduces the OVX-Induced Increase in Bone Marrow Adiposity

Increased bone marrow adiposity has been shown to play a key role in OVX-induced bone loss [27,28]. Therefore, we explored next, whether treatment with OS or HU-671 had any effect on bone marrow fat content. In the representative photomicrographs of the distal femoral metaphysis stained with H&E, bone marrow adipocytes can be clearly seen as transparent ellipsoids in the marrow cavity (Figure 6A). Indeed, adipose tissue occupied a larger percentage of the marrow space in the OVX/Veh-treated group in comparison with the Sham-OVX/Veh-, OVX/OS-, and OVX/HU-671-treated animals. This elevation in fat accumulation in the OVX/Veh-treated animals stemmed from increased adiposity (Figure 6B), adipocyte number (Figure 6C), and adipocyte area (Figure 6D). Remarkably, all of these parameters were normalized following treatment with either OS or HU-671 at 3 mg/kg/day or 0.5 mg/kg/day respectively, indicating a substantial bone marrow anti-adipogenic effect.

### 2.4. HU-671 Has Enhanced Inhibitory Effects on Osteoblast-To-Adipocyte Trans-Differentiation

Mechanistically, the increased bone marrow adiposity can be attributed to the increased trans-differentiation of osteoblasts to adipocytes [29]. To investigate the effect of treatment with OS and its methylated derivative HU-671 on osteoblast-to-adipocyte trans-differentiation, primary calvarial osteoblasts were cultured under adipogenic conditions with and without the addition of the compounds. Measurement of fat accumulation by a fluorescent probe as well as Oil Red O staining indicated that HU-671 had superior inhibitory effects on cellular fat accumulation (Figure 7A,B). Moreover, exposure to HU-671 significantly reduced the expression levels of Rankl, a master regulator of osteoclastogenesis (Figure 7C), and reduced several adipogenic markers, including Pparγ, Cebpα, and Fabp4 (Figure 7D–F), suggesting that HU-671 mitigates the trans-differentiation of osteoblasts-to-adipocytes in the presence of an adipogenic environment.

## 3. Discussion

The present study provides evidence showing that α-methylation of OS specifically potentiates its skeletal in vivo and in vitro efficacies. Of the two OS analogs tested here, only HU-671, and not HU-681, successfully reversed the OVX-induced bone loss at a significantly lower dose in comparison to OS. The restoration of bone mass by HU-671 was due to dual anabolic and anti-catabolic effects and the normalization of bone marrow adiposity. These findings strongly support the documented anti-apoptotic effects of HU-671 [25].

Accumulating evidence demonstrates that fatty acid derivatives play a key regulatory role in a variety of tissues [11]. However, the study of skeletal lipidomics is just emerging and our knowledge of the role of fatty acid derivatives in the control of skeletal remodeling and bone mass is limited. The most prominent compounds investigated are prostaglandins and endocannabinoids [18,30,31]. In previous studies, we demonstrated that several endocannabinoids and endocannabinoid-like compounds are found in bone and that some of them affect the bone remodeling process [6,18,32]. Of those compounds, OS was found to be a potent anti-osteoporotic agent both in vitro and in vivo [24]. Moreover, circulating levels of OS reflect and are affected by bone mass status, indicating its possible role as a skeletal biomarker [24,25]. OS is produced locally in bone and probably in other tissues. Oleic acid may be a precursor for OS, since long-chain fatty acids are known to be metabolized in vivo into amides [33]. Oleic acid is abundantly present in blood and biosynthesized endogenously from stearic acid [34] or provided exogenously. Interestingly, the incidence of osteoporosis is lower in Mediterranean countries, a fact attributed to the high olive oil consumption, which is the richest source of oleic acid and OS [35,36,37,38,39]. Other OS precursors may include *N*-acyl phosphatidylserines [40]. Whether these or other biosynthetic pathways are involved in OS production remains to be established.

OS is a promising compound from a therapeutic point of view; however, with an unprotected amide group, it can be easily hydrolyzed by amidases. While the exact degradation pathway of OS is currently unknown, evidence indicates fatty-acyl amide hydrolase (FAAH), the main endocannabinoid degrading enzyme, as a candidate. Interestingly, FAAH activity is negatively regulated by estrogen [41,42], a possible explanation for the significant reduction of OS levels observed after ovariectomy [24]. Attempting to enhance its potency, novel methylated OS derivatives (HU-671 and HU-681) were prepared and characterized in vivo and in vitro. The substitution of a methyl group adjacent to the amide bond in OS presumably hinders enzymatic hydrolysis of the amide. This assumption is based on extensive published data on a structurally similar compound AEA, whose methylation enhances its metabolic stability and activity [26]. Indeed, our earlier findings indicate that HU-671 and HU-681 exhibit proliferative and mitogenic activity similar to OS in terms of the dose yielding the peak stimulation and magnitude of the stimulatory effect [25]. Interestingly, whereas α-methylation of serine (in HU-671) enhances OS’s anti-osteoclastogenic effect, the methylation of the oleic acid (in HU-681) abolishes it, which could indicate that this modification interferes with OS signaling in osteoclasts and that the effects of OS and its derivatives are cell-type dependent [25]. Despite being structurally similar to AEA, OS does not bind to either cannabinoid receptor 1 or 2 and its putative receptor remains unknown [24]. Identification of the exact receptor for OS is crucially needed to determine its mechanism of action as well as the binding affinities of the synthesized derivatives.

In line with the data in osteoblast and osteoclast cultures, administration of OS and HU-671 to mice commencing 6 weeks after OVX, when bone loss is at a very low rate, if it exists at all, leads to a substantial inhibition of the osteoclast number and stimulation of bone formation. In addition, OS and HU-671 rescued the OVX-induced bone loss in distal femoral metaphysis (~63% rescue by OS and ~65% by HU-671) and in vertebral bodies (~30% rescue for both OS and HU-671) at 3 and 0.5 mg/kg/day, respectively, demonstrating that HU-671 is approximately 10-fold more efficacious than OS. In vitro OS and HU-671 possess a bell-shaped stimulatory dose response curve in which doses at the lower and higher ends yield a lower response [24,25]. In according with this observation, higher doses of both OS and HU-671 were less efficacious at rescuing the OVX-induced bone loss. That bell-shaped effect has also been observed in phytocannabinoids, endocannabinoids, and synthetic-cannabinoids [43,44,45,46]. With that in mind, it is important to note that the dose response effects of both OS and HU-671 were tested in vivo solely in this study, thus further studies should be performed to validate the exact therapeutic window.

As expected of a bone anabolic agent [47], the rescue of bone loss by HU-671 was associated with increased trabecular thickness, trabecular number, and connectivity density. Moreover, both compounds positively influenced serum bone turnover markers by normalizing CTX-1 levels and partially halting the reduction in osteocalcin levels. In this study, as well as in our previous studies, the effect of OS and HU-671 on serum osteocalcin levels was quantified 6 weeks post-treatment [24,25], a time-point at which osteocalcin levels may have already been stabilized [48]. In order to fully assess the effects on osteocalcin levels, a time-line study is required. Altogether, the cumulative effect of the stimulated bone formation and restrained bone resorption lead to the reversal of bone loss to a level insignificantly different from that measured in the Sham-OVX/Veh-treated controls. By comparison, in a similar mouse model, parathyroid hormone 1-34 (PTH (1-34)), a clinically approved bone anabolic agent, rescued only 35% of the OVX-induced bone loss [49]. Although HU-681 also increased the trabecular thickness, its rescue of the OVX-induced bone loss was not enough to reach statistical significance, probably a result of its reduced anti-resorptive ability.

Numerous clinical studies have shown that osteoporosis is associated with increased bone marrow adiposity [50,51,52]. This phenomenon has also been observed in experimental animal models for aging, OVX, glucocorticoid treatments, and diabetes [27,53,54,55,56]. Moreover, excessive infiltration of fat into bone marrow is inversely correlated with bone mineral density values and is considered a therapeutic target for bone loss prevention [57,58,59]. In fact, some of the current anti-osteoporotic drugs, such as Strontium ranelate, Risedronic acid, and Peroxisome proliferator-activated receptor-γ2 antagonist (bisphenol-A-diglycidyl ether) have been shown to lower bone marrow adiposity in human and animal models [60,61,62,63,64]. In the current study, as a result of OVX and estrogen depletion, bone marrow adiposity was significantly elevated, manifested by the increased number (hyperplasia) and size (hypertrophy) of marrow adipocytes. Treatment with OS or HU-671 at 3 mg/kg or 0.5 mg/kg, respectively, normalized all of the bone marrow adiposity parameters. In light of a similar effect by HU-671 in a genetic model for osteoporosis [25], these results further support its therapeutic advantage.

Osteoblasts and adipocytes originated from common mesenchymal progenitor cells (MSCs). Modulation of the differentiation of the two lineages is influenced by different transcription factors [65]. Changes in the differentiation potential of MSCs, favoring adipogenesis and inhibiting osteoblastogenesis under pathological conditions, such as aging and OVX, have been observed both in vitro and in vivo [66,67,68]. Moreover, bone marrow MSC-derived osteoblasts have the potential to transdifferentiate directly to adipocyte lineage under certain conditions, which may contribute to the increased adiposity observed in osteoporosis [69]. To assess the effects of OS and HU-671 on osteoblast-to-adipocyte trans-differentiation, newborn mouse calvarial osteoblasts (NeMCO) were grown for 21 days under adipogenic conditions. Both OS and HU-671 at 10^−12^ M suppressed lipid accumulation and adipogenesis. Additionally, both treatments reduced the expression levels of *Cebpa* and *Fabp4*, indicating reduced adipogenic differentiation. Interestingly, exposure to HU-671, but not to OS, reduced the expression levels of *Rankl*, which further supports its anti-osteoclastogenic effects. These results are in line with the documented role of HU-671 in inhibiting the trans-differentiation of osteoblasts-to-adipocytes, promoting osteoblastogenesis, and enhancing matrix mineralization previously observed in a genetic model for osteoporosis [25].

## 4. Materials and Methods

### 4.1. Methylated OS Derivatives

OS, as well as its methylated derivatives, oleoyl α-methyl serine (HU-671) and 2-methyl oleoyl serine (HU-681), were synthesized as described previously [25].

### 4.2. Animals and Experimental Protocol

The experimental protocol used was approved by the Institutional Animal Care and Use Committee of the Hebrew University of Jerusalem, which is an Association for Assessment and Accreditation of Laboratory Animal Care (AAALAC) International accredited institute. C57BL/6J mice were used in all experiments. Female, 8-week-old mice were subjected to bilateral OVXor sham-OVX; 6 weeks later, mice were administered intraperitoneally daily injections of OS, HU-671, HU-681, or vehicle (ethanol/emulphore/saline (1:1:18)) for 6 weeks. Based on the well-established efficacy of OS in this model, and the previously reported efficacy of HU-671 and HU-681 [24,25], the following in vivo doses were used: OS: 1, 3, or 9 mg/kg/day; HU-671: 0.1, 0.5, or 1 mg/kg/day; HU-681: 0.3, 1, or 5 mg/kg/day. To assess the in vivo effect of the drugs on bone formation, newly formed bone was vitally labeled by fluorochrome calcein (Sigma-Aldrich, MO, USA) that was injected intraperitoneally (15 mg/kg) 4 days and 1 dday before euthanization. Mice were euthanized by a cervical dislocation under anesthesia. Once euthanized, trunk blood was collected for determining the biochemical parameters, and the femoral bones and L3 vertebrae were separated, cleaned, and fixed in 10% phosphate buffered formalin (pH 7.2) for 48 h, and then kept in 70% ethanol until further use.

### 4.3. The Effects of Methylated OS Derivatives on Bone Structure and Remodeling

The skeletal activity of OS and its methylated derivatives was assessed by combined micro-computed tomography (μCT)/histomorphometric analyses as described previously [25]. Briefly, femora and L3 lumbar vertebrae were examined by a μCT system (μCT 40; Scanco Medical AG) at 10-μm resolution in all three spatial dimensions. In the femora, trabecular bone parameters were measured in a metaphyseal segment, extending proximally from the proximal tip of the primary spongiosa to the proximal border of the distal femoral quartile. Cortical bone parameters were determined in a diaphyseal segment extending 1.12 mm distally from the midpoint between the femoral ends. Trabecular bone parameters were also analyzed in L3 bodies. After μCT image acquisition, the femoral specimens were embedded undecalcified in polymethylmethacrylate (Technovit 9100; Heraeus Kulzer, Wehrheim, Germany). Longitudinal sections through the midfrontal plane were left unstained for dynamic histomorphometric analyses, based on the vital calcein double labeling. To identify osteoclasts, consecutive sections were deplasticized and stained for tartrate-resistant acid phosphatase (TRAP; Sigma-Aldrich, MO, USA), and counterstained with Mayer’s hematoxylin. Histomorphometric analysis was carried out on digital photomicrographic images by using IMAGE-PRO PLUS V.6 image analysis software (Media Cybernetics, MD, USA). The following parameters were determined: the mineral appositional rate (MAR), mineralized surface (MS/BS), bone formation rate (BFR/BS), and osteoclast number (N.Oc/BS). The terminology and units used for these measurements were according to the convention of standardized nomenclature [70].

### 4.4. Cell Culture

NeMCO were prepared from 4 to 5-day-old mice by successive collagenase digestion as described previously [71]. The cells were grown to confluence in α-MEM supplemented with 10% fetal calf serum (FCS), and then plated in 6-well plates for RNA extraction, in 12-well plates for Oil Red O staining, and in 96-well plates for AdipoRed labelling. Induction of trans-differentiation of osteoblasts to adipocytes was achieved by using an adipogenic induction kit (hMSC Adipogenic Differentiation Medium BulletKitTM; Lonza, Basel, Switzerland). Cells were grown according to the manufacturer’s instructions. Throughout the experiment, OS and HU-671 were added to the growth medium for 21 days at a concentration of 10^−12^ M to assess their effect on adipogenic differentiation.

### 4.5. AdipoRed Labelling

Intracellular lipid accumulation was quantified using AdipoRed Adipogenesis Assay Reagent (Lonza, Basel, Switzerland) according to the manufacturer’s protocol. Briefly, cells were washed once with 1 × PBS and incubated with AdipoRed Reagent for 10 min. Fluorescence was detected at an excitation of 485 nm and an emission of 572 nm, using a microplate reader at 37 °C.

### 4.6. Oil Red O Staining

Staining was performed using 0.21% Oil Red O in 100% isopropanol (Sigma-Aldrich, MO, USA). Briefly, cells were fixed in 10% formaldehyde, stained with Oil Red O for 10 min, and rinsed with 60% isopropanol (Sigma-Aldrich, MO, USA). Oil Red O was eluted by adding 100% isopropanol for 10 min and the optical density (OD) measured at 490 nm.

### 4.7. Serum Markers of Bone Remodeling

Serum osteocalcin levels were determined using a two-site EIA kit (Biomedical Technologies Inc., Lancashire, UK). Mouse C-telopeptide of type I collagen (CTX-1) was measured in the same specimens using an EIA kit (Wuhan EIAab Sciences Co., Ltd., Wuhan, China).

### 4.8. Real-Time PCR

Total mRNA from cell cultures was extracted using Bio-Tri RNA lysis buffer (Bio-Lab, Jerusalem, Israel), followed by DNase I treatment (Thermo Scientific, IL, USA), and then reverse transcribed using the Iscript cDNA kit (Bio-Rad, CA, USA). Real-time PCR was performed using the iTaq Universal SYBR Green Supermix (Bio-Rad, CA, USA) and the CFX connect ST system (Bio-Rad, CA, USA). Primers are listed in Table 1.

### 4.9. Statistical Analysis

Values are expressed as the means ± SEMs. Statistical analyses were performed with GraphPad Prism 6.0 (GraphPad Software, CA, USA). Data were analyzed by ANOVA with Tukey’s or Dunnet’s multiple comparison tests. Significance was set at *p* < 0.05.

## 5. Conclusions

In summary, this study suggests that α-methylation hinders OS’s metabolism, thus enhancing its in vitro and in vivo effects by extending its availability to its target cells. In addition, the present data provide a preclinical proof for further development of HU-671-based anti-osteoporotic therapy. The potential advantages of such therapy are the concomitant bone anabolic and anti-resorptive activities, and the anti-adipogenic effects.

## 6. Patents

The use of OS and its methylated derivatives for bone diseases is protected by US Granted Patent number 12/936,498, as well as US PCT Patent number 62/322,555.

## Figures and Tables

**Figure 1 molecules-24-03719-f001:**
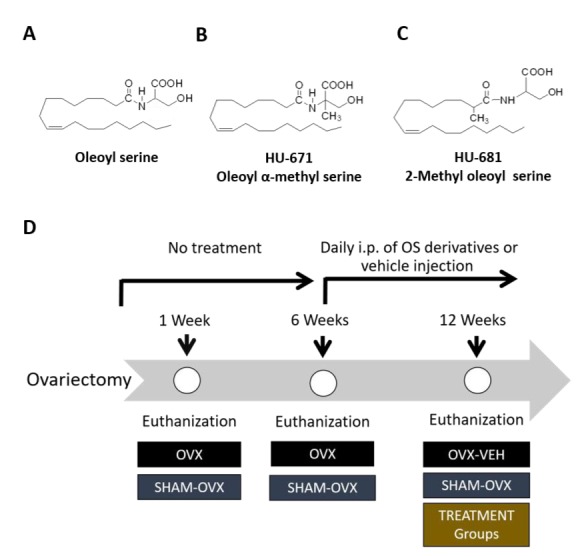
Experimental design to test the efficacy of methylated OS derivatives in a rescue model for osteoporosis. The chemical structure of OS (**A**) and its methylated derivatives, oleoyl α-methyl serine (HU-671) (**B**), and 2-methyl oleoyl serine (HU-681) (**C**). Schematic representation of the experimental design for testing the rescue of bone loss by OS, HU-671 and HU-681; 8 week-old female mice were either OVXed or sham-OVXed and analyzed one week, and 6 weeks after surgery to serve as controls. 6 weeks post operation, daily intraperitoneal treatment with Vehicle, OS, HU-671, or HU-681 was initiated for an additional 6 weeks (**D**).

**Figure 2 molecules-24-03719-f002:**
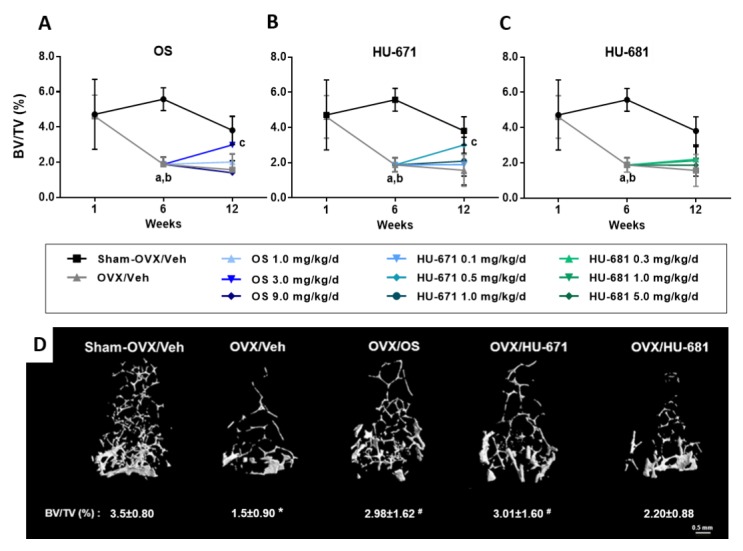
Assessment of trabecular bone volume density in the distal femoral metaphysis of ovariectomized (OVXed) mice treated with OS and its methylated derivatives. The trabecular bone volume density (BV/TV) of distal femoral metaphysis after treatment with OS (**A**), HU-671 (**B**), or HU-681 (**C**). Representative 3D images of the distal femoral metaphysis of mice with median BV/TV values for each treatment (**D**). Data represent the means ± SDs obtained from 7–9 mice per condition. ^a^
*p* < 0.05 versus the 1-week OVX/Veh-treated group, ^b^
*p* < 0.05 versus the 6-week Sham-OVX/Veh-treated group, ^c^
*p* <0.05 versus the 12-week OVX/Veh-treated group, * *p* < 0.05 versus the Sham-OVX/Veh-treated group, ^#^
*p* < 0.05 versus the OVX/Veh-treated group.

**Figure 3 molecules-24-03719-f003:**
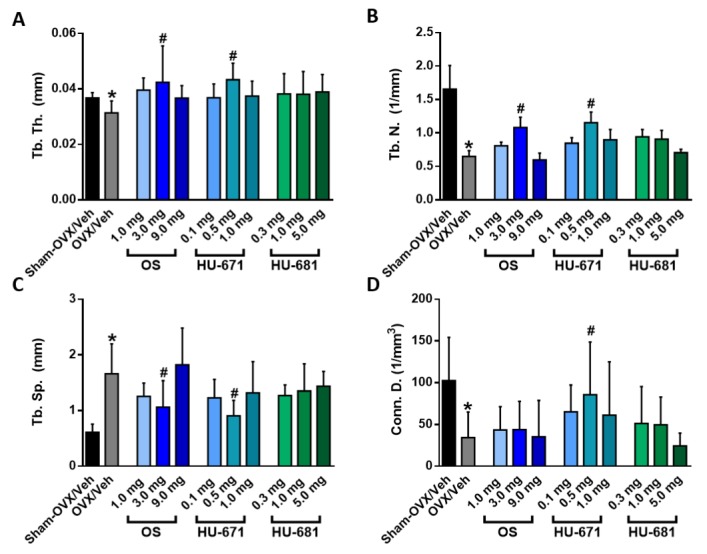
Microstructural analysis of trabecular bone parameters in the distal femoral metaphysis of OVXed mice treated with OS and its methylated derivatives. Trabecular thickness (Tb. Th.) (**A**). Trabecular number (Tb. N.) (**B**). Trabecular spacing (Tb. Sp.) (**C**). Connectivity density (Conn. D.) (**D**). Data represent the means ± SDs obtained from 7–9 mice per condition * *p* < 0.05 versus the Sham-OVX/Veh-treated group, ^#^
*p* < 0.05 versus the OVX/Veh-treated group.

**Figure 4 molecules-24-03719-f004:**
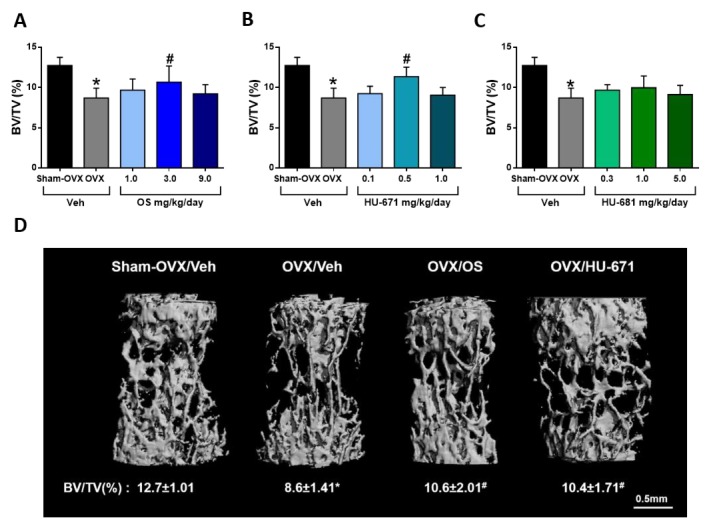
Microstructural analysis of trabecular bone in L3 vertebral bodies of OVXed mice treated with OS and its methylated derivatives. L3 trabecular bone volume density (BV/TV) values of OS- (**A**), HU-671- (**B**), and HU-681- (**C**) treated animals. Representative 3D images of the L3 vertebrae of mice with median BV/TV values for each treatment (**D**). Data represent the means ± SDs obtained from 7–9 mice per condition. * *p* < 0.05 versus the Sham-OVX/Veh-treated group, ^#^
*p* < 0.05 versus the OVX/Veh-treated group.

**Figure 5 molecules-24-03719-f005:**
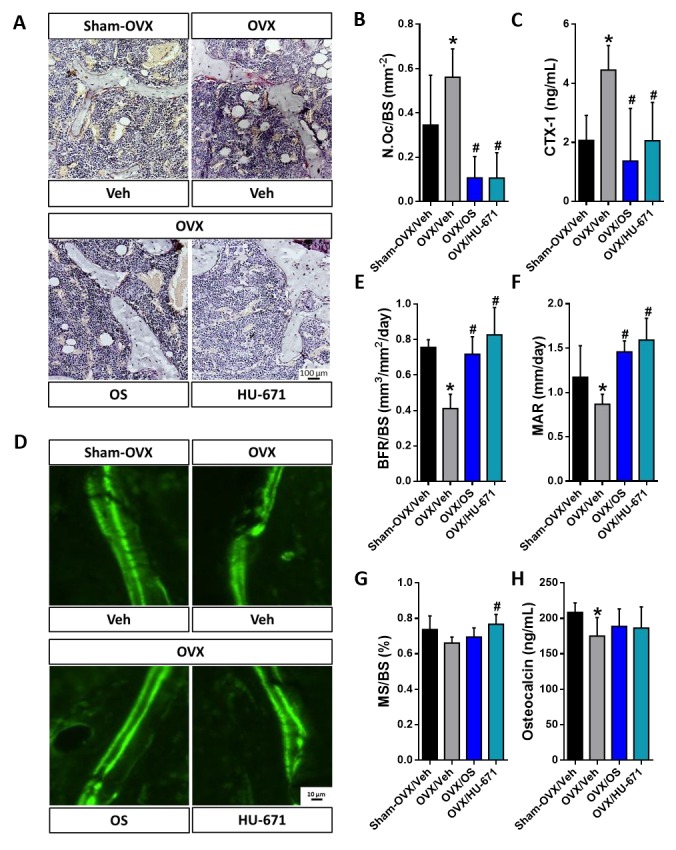
Both OS and HU-671 have bone anabolic and anti-catabolic effects in vivo. Both OS (3 mg/kg/day) and HU-671 (0.5 mg/kg/day) reduced osteoclastogenesis, as measured by the reduction of TRAP+ osteoclasts per trabecular surface area (NOc/BS) (**A**,**B**) as well as normalization of serum CTX-1 levels (**C**). Similar improvements in the bone formation rate (BFR/BS) (**E**) and the mineral apposition rate (MAR) (**D**,**F**) were found in OVXed mice treated with either OS or HU-671. The mineralized surface (MS/BS) was increased only in HU-671-treated OVXed mice (**G**). Both drugs were not effective at fully reversing the reduction in serum osteocalcin levels (**H**). Data represent the means ± SDs obtained in 4–8 mice per condition. * *p* < 0.05 versus the Sham-OVX/Veh-treated group, ^#^
*p* < 0.05 versus the OVX/Veh-treated group.

**Figure 6 molecules-24-03719-f006:**
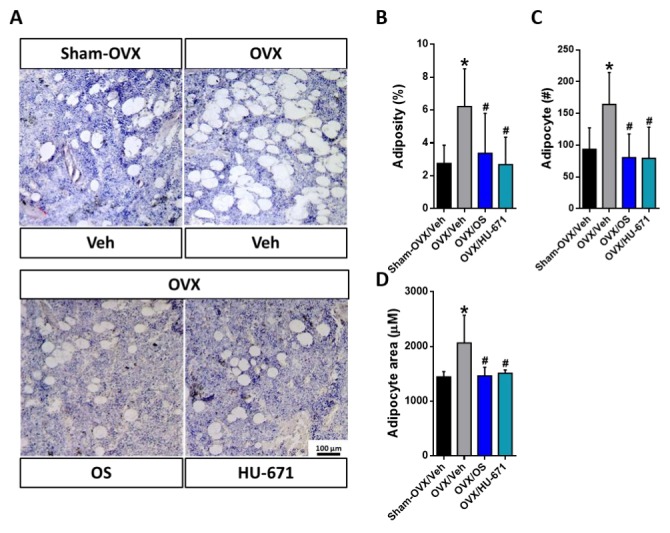
Reversal of bone marrow adiposity by OS and HU-671. Representative images of bone marrow adiposity in H&E stained specimens of control and treatment groups (OS, 3 mg/kg/day; HU-671, 0.5 mg/kg/day; **A**), adiposity percentage (**B**), the number of adipocytes (**C**), and adipocyte area (**D**). Data represent the means ± SDs obtained in 5–8 mice per condition. * *p* < 0.05 versus the Sham-OVX/Veh-treated group, # *p* <0.05 versus the OVX/Veh-treated group.

**Figure 7 molecules-24-03719-f007:**
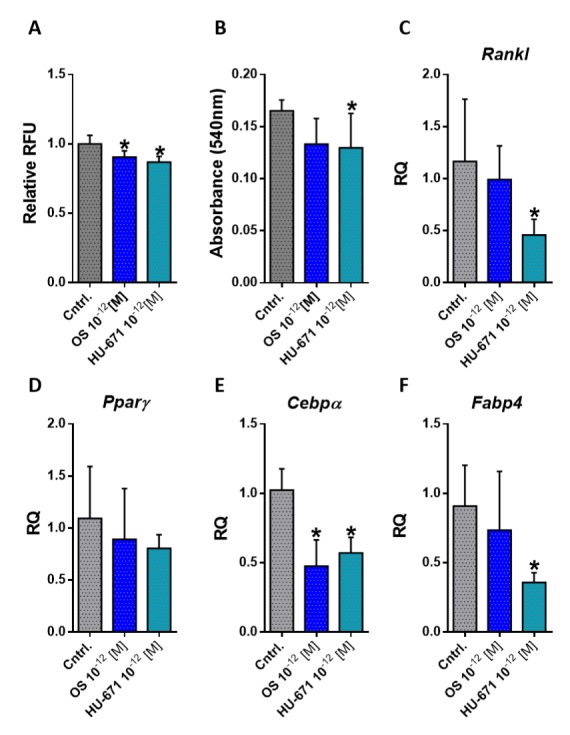
HU-671 exhibits enhanced anti-adipogenic effects. The inhibition of fat accumulation in primary calvarial osteoblasts cultured under adipogenic conditions by OS and HU-671 was quantified by AdipoRed (**A**) and Oil Red O staining (**B**). The altered expression profiles of the receptor activator of nuclear factor kappa-B ligand (Rankl) (**C**), peroxisome proliferator-activated receptor gamma (Pparγ) (**D**), CCAAT/enhancer binding protein alpha (Cebpα) (**E**), and fatty acid binding protein 4 (Fabp4) (**F**) were attenuated following chronic (21 days’) in vitro exposure to HU-671. Data represent the means ± SDs obtained in 6–12 replicates per condition. * *p* < 0.05 versus the Veh-treated controls.

**Table 1 molecules-24-03719-t001:** Real-Time PCR Primer Sequences.

	FORWARD	REVERSE
**Mus musculus *Gapdh***	ACCAGGGAGGGCTGCAGTCC	TCAGTTCGGAGCCCACACGC
**Mus musculus *Cebpα***	CAAGAACAGCAACGAGTACCG	GTCACTGGTCAACTCCAGCAC
**Mus musculus *Fabp4***	AAGGTGAAGAGCATCATAACCCT	TCACGCCTTTCATAACACATTCC
**Mus musculus *Pparγ***	TCGCTGATGCACTGCCTATG	GAGAGGTCCACAGAGCTGATT
**Mus musculus *Rankl***	TCCAGCTATGATGGAAGGCT	GTACCAAGAGGACAGAGTG

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
