# Peer review of "HU-671, a Novel Oleoyl Serine Derivative, Exhibits Enhanced Efficacy in Reversing Ovariectomy-Induced Osteoporosis and Bone Marrow Adiposity"

_molecules, 2019, doi:10.3390/molecules24203719_

Round 1
Reviewer 1 Report
It is an interesting paper that bring some new data about the effects of a oleoyl serine derivative as anti-osteoporotic substance.
Major points:
The finding that serum levels of osteocalcin were not significantly changed in mice treated with either 144 OS or HU-671 should be discussed.
HU-671 was about 10-fold more efficacious than OS in OVX-induced bone loss. Why higher OS or HU-671 concentrations had no effect in trabecular bone volume density (Figure 2)? Moreover, why treatment with HU-681had no effect? These findings should be discussed.
Conclusions and Abstract: The authors stated that this study suggests that α-methylation interferes with amidase activity. However, no studies were performed that could show such interference. In my opinion what would be appropriate would be to postulate that methylation would prevent OS metabolization by amidase. If the authors agree with this suggestion, possible mechanisms should be discussed in Discussion section.
Minor points
The schematic representation of the experimental design to test the efficacy of methylated OS derivatives shown in Figure 1D seems to be different from that stated in text. In the figure it seems that treatments were initiated 12 weeks after ovariectomy, while in the text after 6 weeks.
Figure legends (5 and 6) should contain information regarding the OS and HU-671 concentrations tested.
Reviewer 2 Report
The paper by Baraghithy et al. presents the use of a novel Oleoyl Serine Derivative to rescue bone loss in ovariectomized mice. The experimental design is appropriate and the results of potential clinical interest.
Some minor concerns follow:
1) Did the authors documented any side effect of HU-671 compound (i.e. changes in blood count, survival)?
2) Are there changes in post-ovariectomized bone OS levels?
3) Greater magnification of panel 5A should be provided.
4) Together with the anti-adipogenic effects, it would be of great interest to show also the effect of HU-671 on osteogenic e mitogenic activity of osteoblasts.
5) All statistics are presented as mean±SEM. It would be much more appropriate to use graphs showing dispersion of the data or use standard deviation.
6) Can the authors discuss about the evidence that increased concentration of the compound exerts lower effect?
Round 2
Reviewer 1 Report
The authors have adequately answered the questions and changed the text according to the suggestions.